# ONE TRAINING FITS ALL: GENERALIZED DATA CONDENSATION VIA MIXTURE-OF-INFORMATION BOTTLENECK GUIDANCE

## ABSTRACT

Data condensation (DC) technologies are widely used in buffer-constrained scenarios to reduce the memory demands of training samples while maintaining the training performance of deep neural networks. However, due to the storage constraints of deployment devices and the high energy costs associated with the condensation process, synthetic datasets generated by DC often suffer from inferior performance in terms of training efficiency and scalability, which significantly limits their practical application across various edge devices. This issue arises from two main factors: (i) existing state-of-the-art (SoTA) data condensation approaches update synthetic datasets by intuitively matching intermediate training outputs (e.g., gradients, features, and distributions) between real and synthetic datasets, without enhancing their representational information capabilities from the perspective of the useful information contained; (ii) DC methods do not adequately consider the heterogeneity of storage constraints across different edge devices, leading to excessive training overheads (i.e., increased consumption or storage requirements). To address these challenges, we propose a novel method named Mixture-of-Information Bottleneck Dataset Condensation (MIBDC), which employs information bottlenecks from synthetic datasets with varying Images Per Class (IPC) to enhance overall DC generalization and scalability. In particular, this paper identifies the following two phenomena: (i) the quality of synthetic datasets improves with an increase in synthetic dataset quantity, and (ii) the smaller the synthetic dataset, the earlier it reaches the convergence peak. Based on these findings, this paper proposes that (i) larger synthetic datasets can guide the more effective convergence of smaller ones, and (ii) the information contained in synthetic datasets with different IPC numbers can collaboratively enhance dataset condensation generalization. Comprehensive experimental results on three well-known datasets demonstrate that, compared with SoTA dataset condensation methods, MIBDC not only improves the generalization performance of trained models but also decreases training times for various edge devices (**training once**).

## 1 INTRODUCTION

Along with the development of artificial intelligence, data-driven Deep Neural Networks (DNNs) have been widely used in storage resource-limited scenarios such as autonomous driving (Wen et al.), industrial control, robotics and smart healthcare. However, since DNN training requires sufficient resources, the performance of DNNs will be severely limited if the quality and quantity of training samples cannot be guaranteed. For example, the CLIP or Large Language Model consumes significant storage resources for its training samples and requires hundreds of GPU hours to achieve an acceptable performance. To address such increasing storage requirements of training datasets and to accommodate the limited memory capabilities of various Internet of Things (IoT) devices, Dataset Condensation (DC) technology has been proposed to ensure that effective information contained in large training datasets can be condensed into a small synthetic dataset. In DC, the key problem is how to effectively explore and condense proper information from real datasets into corresponding synthetic datasets without degrading their representation information capabilities.

To solve this problem, many works are proposed, which can be mainly divided into two classes, i.e., parameter matching methods and distribution matching methods, as shown in Figure 1. The former is proposed to find superior parameters to align during the DNN training procedure, thus

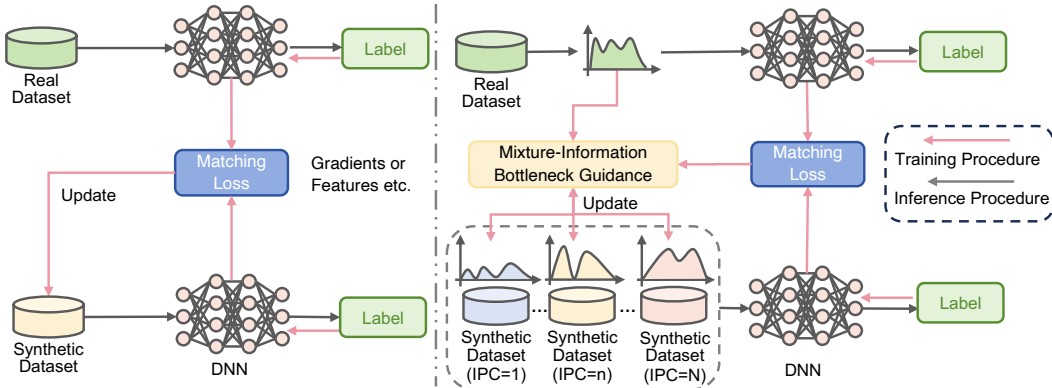

Figure 1: (Left) General framework of previous SoTA DC methods: matching intermediate products extracted by networks from real and synthetic datasets; (Right) the framework of MIBDC: maximizing the mutual information based on various datasets (including real dataset and label).

optimizing the representation information capability of the synthetic dataset. For example, Zhao et al. (2021) used gradient matching between the same batch from synthetic datasets and real datasets with the same DNN. Moreover, Wang et al. (2022) optimized the feature map between the real datasets. Liu et al. (2023) used the combination to optimize the distance, and Shang et al. (2024) used mutual information to calculate the distances between synthetic datasets and real datasets. However, since parameters occupy a very large amount of storage, this requires a huge amount of memory space during the training procedure. The latter is proposed to enhance the representation capability of synthetic datasets by the intermediate outputs (i.e., dataset distribution) of DNNs. The above two kinds of methods can partially improve the representation capability of synthetic datasets, which inevitably yields the problem of a trade-off between representation capability and limited storage. This dilemma arises because the information representation capabilities of synthetic datasets are inherently bottlenecked, which greatly limits the training capabilities of datasets with low IPC numbers. Therefore, *how to squeeze effective information into multi-smaller scales synthetic datasets and preserve as much useful information as possible from the perspective of information contained with superior scalability is becoming one of the most urgent challenges in the development of DC.*

To solve the above issues, we find that different numbers of synthetic datasets can be treated as variants containing different effective information in terms of DNN training performance, where the more information contained, the better DNN obtained. Based on these findings, inspired by the method of Information Bottleneck (IB), we introduce this metric in synthetic datasets, which can not only optimize and explain how to extract and transfer effective information in dataset condensation but also naturally capture the non-linear statistical dependence between real datasets and synthetic datasets. Inspired by the information contained in the synthetic dataset, we propose a novel method called mixture-of-information bottleneck guidance in DC. Specifically, we define the knowledge distillation problem as a bidirectional optimization problem, involving maximizing the mutual information between real data sets and synthetic data sets and minimizing the mutual information between synthetic data sets and labels. Then, we derive a mixture-of-information bottlenecked method for the dataset condensation task using various synthetic datasets in the MI bi-directional optimization problem of DC. Finally, we design a highly effective optimization strategy in a collaborative manner for the DC task using the proposed Mixture-of-Information Bottleneck (MIB) methods for Mutual Information (MI) maximization. In this way, the mixture-information bottleneck using a collaborative paradigm can not only lead to MI maximization but also minimize the MI from the label. To the best of our knowledge, it is the first work aiming at optimising the information bottleneck of synthetic datasets during the DC within a collaborative learning paradigm.

Overall, the contributions of this paper are three aspects. i) To condense the effective information from a large real dataset to a smaller synthetic dataset under an efficient-reasonable validation metric, we formulate the DC as an MI bi-directional optimization within multi-information bottleneck constraints (i.e., real data, other IPC synthetic data and its corresponding labels). To the best of our knowledge, this is the first work to introduce IB into the DC domain. ii) To optimize the defined problem in

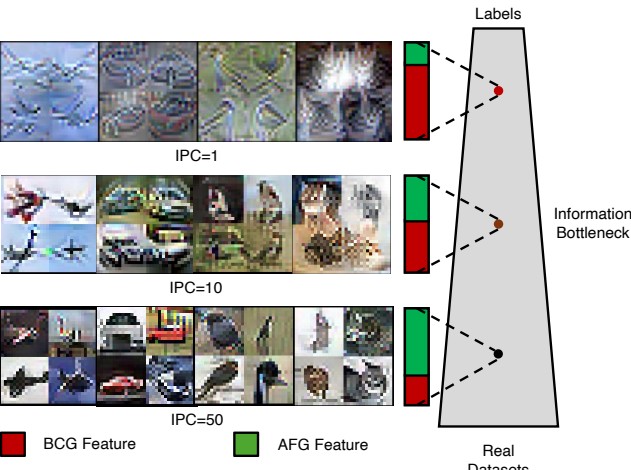

Figure 2: Motivation of our mixture-of-information bottleneck guidance method.

DC, we derive a highly effective mixture-information bottleneck-guided method. In this way, invalid information irrelevant to labels will be omitted during optimization, while key information relevant to labels will be retained in the synthetic dataset. iii) Comprehensive experiments on four well-known datasets show that our method outperforms existing SoTA methods in the DC domain.

## 2 MOTIVATION

In the motivation for this paper, as depicted in Figure 2, we observed that during the training procedure, synthetic datasets with a smaller IPC intend to converge more quickly, whereas those with a larger IPC converge more slowly. Specifically, the synthetic dataset with an IPC of 1 reaches convergence in the early stages, while the dataset with an IPC of 50 converges much later. This observation means that datasets with a smaller IPC can learn features that are simpler with low-density information and more directly related to the label, whereas datasets with a larger IPC tend to learn features with high-density information that are indirectly related to the label. Therefore, we assume that there are two classes of features in the synthetic datasets (i.e., *Basic-Coarse-Grained features* and *Advanced-Fine-Grained features*) after the dataset condensation procedure. The former features represent basic features of the real datasets that provide an overview but miss more dataset details. Thus, during DC, the smaller IPC in a synthetic dataset can quickly obtain these features that are easier to obtain but may miss some fine and intricate details. On the contrary, the latter feature is more challenging to acquire but contains richer, more dense information that can be helpful in DNN learning. So, the larger IPC dataset can not only learn the above basic information but also capture detailed, in-depth features that offer a comprehensive understanding of the real dataset. Thus, we think that during the dataset condensation procedure, there should be an information-based transformation between Basic-Coarse-Grained (BCG) features and Advanced-Fine-Grained (AFG) features. For synthetic datasets with higher IPC, the contained AFG features tend to overlap with BCG features, preventing the model from effectively learning this high-level information. We assume that this occurs because BCG features are more closely associated with the label in the method of information bottleneck. Consequently, during supervised learning, once the model learns the BCG features, it does not learn the AFG features, leading to insufficient accuracy in synthetic datasets with lower IPC.

Based on the above assumptions, to better optimize the above objectives, we introduce a multi-synthetic dataset engaged in a collaborative learning framework in DC, named mixture-of-information bottleneck guided dataset condensation, which is based on IB of various multi-sized synthetic datasets. Based on the information bottleneck method, the mixture-of-information bottleneck is similar to the knowledge distillation method, whose main principle is to select a suitable teacher for learning tasks to expand the contained information of the synthetic dataset in AFG features.

## 3 METHOD

In this section, we introduce the methodology of Mixture-of-Information Bottleneck guidance for Dataset Condensation (MIBDC). First, this paper introduces the existing SoTA DC method and then introduces the background and effect of the information bottleneck. Later, this paper formalizes the DC problem as an information compression problem. Then, we transform the current data compression target into a learnable target optimization problem and introduce our mixture-of-information bottleneck loss to solve this problem. Finally, we discuss the feasibility and theoretical proof of the method and give a corresponding discussion.

### 3.1 PRELIMINARIES

#### 3.1.1 DATASET CONDENSATION

The goal of dataset condensation is to generate a small synthetic dataset $D^*_{syn} = \{X_{syn}, Y_{syn}\}$, where the same architecture model can be trained obtaining a comparable performance relative to the real dataset. In this way, the training time of neural networks will be greatly reduced based on such small synthetic datasets. Based on this method, some representative DD methods generally recommend that similar intermediate outputs be enforced when training models on real and synthetic datasets. Existing methods can be formalized as the following optimization problem:

$$D^*_{syn} = \arg\min_{D_{syn}} \mathbb{E}_{X_{syn} \sim D_{syn}} \mathbb{E}_{X_r \sim D_{real}} \left[ \text{Dist} \left( f(X_{real}; \theta^*), f(X_{syn}; Y_{syn}) \right) \right]. \tag{3.1}$$

Here, $\theta$ are the learned weights of networks trained on the real dataset $D_{real}$ and the synthetic datasets. The distance function Dist can be used to calculate the distance between the real and synthetic datasets for obtaining synthetics datasets. By aligning the two extracted intermediate results, one can optimize the synthetic data to achieve comparable performance to the real data set. Note that how to design the distance function alignment of networks trained on different datasets to optimize synthetic datasets is the key in previous DD methods, such as Lee et al. (2022); Wang et al. (2018); Zhao et al. (2021). Although these distance metrics of DC design bring relatively satisfactory results, since the data set compression itself is a data compression problem, how to better increase the effective information of the compressed data set and reduce the useless information of the compression into a synthetic data set has never been considered.

#### 3.1.2 INFORMATION BOTTLENECK

As a promising method in information extraction, Information Bottleneck (IB) is proposed to extract relevant information about a target variable $Y$ from an input variable $X$. This method compresses $X$ into a more efficient variable $T$, maximizing the relevant information about $Y$ retained. It balances the compression and retention of information by minimizing $I(T; X)$(the complexity of compression) and maximizing $I(T; Y)$(i.e., the information about $Y$ preserved.) The trade-off is regulated by the parameter $\beta$, a Lagrange multiplier in the optimization formula:

$$\max_{\mathbb{P}(t|x)} \{I(X; T) - \beta I(T; Y)\}. \tag{3.2}$$

Here, $\mathbb{P}(t|x)$ is the conditional probability distribution of $T$ given $X$, optimized to finely balance between compressing $X$ and retaining essential information about $Y$. Mutual Information (MI), denoted as $I(X, Y)$, quantifies the information gained about one random variable by comparing another. It is measured in bits and reflects the reduction in uncertainty about one variable when the other is known. High mutual information indicates a significant reduction in uncertainty, and vice versa. Strictly, for two discrete variables $X$ and $Y$, their MI can be defined as:

$$I(X, Y) = \sum_{x,y} \mathbb{P}_{XY}(x, y) \log \left( \frac{\mathbb{P}_{XY}(x, y)}{\mathbb{P}_X(x)\mathbb{P}_Y(y)} \right), \tag{3.3}$$

where $\mathbb{P}_{XY}(x, y)$ is the joint distribution, $\mathbb{P}_X(x) = \sum_y \mathbb{P}_{XY}(x, y)$ and $\mathbb{P}_Y(y) = \sum_x \mathbb{P}_{XY}(x, y)$ are the marginals of $X$ and $Y$, respectively. Under this constraint, the dataset condensation problem's target is to save effective information from real datasets and, like the data compression problem, under the constraints in information bottleneck.

## 3.2 INFORMATION BOTTLENECK ESTIMATION

In this section, we first formalize the current dataset condensation problem and propose two optimization objectives: First, the mutual information between synthetic and real datasets is maximized, and second, the mutual information between synthetic datasets and labels is minimized to enhance the generalization in the DC procedure. Then, based on such objectives, we propose a mixture-of-information bottleneck guidance method that utilizes synthetic datasets with various IPC numbers to guide the DC generalization. From the perspective of information bottleneck, the real and synthesized data with lower IPC numbers can be compressed more compactly and save more effective AFG features by using IB guidance.

**Problem Formulation**: Actually, for variable $\mathbf{X}_{\text{real}}$ representing the samples in the real dataset and $\mathbf{X}_{\text{syn}}$ means the one in synthetic datasets, we want to maximize the MI between $\mathbf{X}_{\text{real}}$ and $\mathbf{X}_{\text{syn}}$ and minimize the MI between the $\mathbf{X}_{\text{real}}$ and real label Y, i.e.,

$$\text{Optimization Objective:} \quad \mathbf{X}_{\text{syn}}^{\star} = \arg\max_{\mathbf{X}_{\text{syn}}} \left[ I(\mathbf{X}_{\text{real}}, \mathbf{X}_{\text{syn}}) - \beta I(\mathbf{X}_{\text{syn}}, \mathbf{Y}) \right]. \tag{3.4}$$

Note that in this paper, our goal is not direct dataset compression but rather the optimization of a synthetic dataset with a small IPC number. Therefore, $\mathbf{X}_{\text{real}}$ in the IB represents the real dataset, and $\mathbf{Y}$ denotes to its corresponding label. This means that our method employs the IB theory to dataset distillation, by reducing the BCG features in the synthetic dataset and increasing the AFG features, thus improving the generalization performance of the synthetic dataset with a small IPC number for efficient dataset compression. In this way, real dataset $\mathbf{X}_{\text{real}}$ can distil the maximal effective information into the synthetic dataset $\mathbf{X}_{\text{syn}}$. We define the Multi-Layer Perceptron (MLP) in the form of a one-layer MLP as the mutual information estimator to calculate the mutual information, which can be defined as:

$$f(\mathbf{W}; \mathbf{x}) = (\mathbf{W} \cdot \sigma \cdot \mathbf{x}), \tag{3.5}$$

where $\mathbf{x}$ is the input sample and $\mathbf{W} : \mathbb{R}^{d_I} \mapsto \mathbb{R}^{d_O}$ stands for the weight matrix connecting the first and the last layer, with $d_I$ and $d_O$ representing the sizes of the input and output of the last network layer, respectively. The $\sigma(\cdot)$ function performs element-wise activation operations on the input feature maps. Based on those predefined notions, the mutual information estimator $f(\mathbf{x})$ employs the predictions can be obtained by:

$$\mathbf{o}_{\text{syn}}^{j} = f(\mathbf{x}_{\text{syn}}^{j}), \quad j \in \{1, \dots, N\}, \quad \mathbf{o}_{\text{real}}^{j} = f(\mathbf{x}_{\text{real}}^{j}), \quad j \in \{1, \dots, N\}, \tag{3.6}$$

where $N$ is the number of the training samples.

**Theorem 1 (In-variance of Mutual Information):** Mutual information is invariant under the reparametrization of the marginal variables. If $X' = F(X)$ and $Y' = G(Y)$ are homeomorphisms (*i.e.*, $F(\cdot)$ and $G(\cdot)$ are smooth uniquely invertible maps), then

$$I(X, Y) = I(X', Y'). \tag{3.7}$$

Since each information estimator $\mathbf{W} : \mathbb{R}^{d_I} \mapsto \mathbb{R}^{d_O}$ can be considered as the smooth uniquely invertible maps **Theorem 1**. Combining this theorem with the definition of MI in Eq 3.7, we observe that the MI in the targeted data level is equivalent to MI in the predictions level, *i.e.*, ,

$$I(\mathbf{X}_{\text{real}}, \mathbf{X}_{\text{syn}}) = I(\mathbf{O}_{\text{real}}, \mathbf{O}_{\text{syn}}). \tag{3.8}$$

The proof of this theorem are in Shang et al. (2024). Based on the above method, we can obtain MI estimates between real and synthetic datasets and MI estimates between synthetic datasets and labels. In other words, Theorem 1 helps us to better ensure that the mutual information we calculate using intermediate results or predictions is fully theoretically guaranteed. Therefore, we can convert the optimization objective Eq 3.9 to the following target.

$$\text{Estimated Optimization Objective:} \quad \arg\max_{\mathbf{X}_{\text{syn}}} \left[ I(\mathbf{O}_{\text{real}}, \mathbf{O}_{\text{syn}}) - \beta I(\mathbf{O}_{\text{syn}}, \mathbf{Y}) \right]. \tag{3.9}$$

### 3.2.1 MUTUAL INFORMATION NEURAL ESTIMATOR (MINE)

Due to the the mutual information can not be directly calculated Kullback (1997), to tackle this issue, we introduce the MLP-based MINE Belghazi et al. (2018) to estimate the mutual information between two synthetic datasets. After obtaining the above optimization objectives, MINE is employed to

estimate the mutual information $I(\mathbf{X}, \mathbf{Y})$ between two variables $\mathbf{X}$ and $\mathbf{Y}$. Specifically, given the predictions $\mathbf{X}$, the target value $\mathbf{Y}$, and the information estimator $MLP$, we calculate the variational lower bound of the mutual information between $\mathbf{X}$ and $\mathbf{Y}$ as follows: The target value $\mathbf{Y}$ can be randomly shuffled to form a new edge distribution $\tilde{\mathbf{Y}} = \text{random}(\mathbf{Y})$. Secondly, the prediction of the joint and marginal distributions can be defined as follows:

$$(x^{(1)}, y^{(1)}), \ldots, (x^{(b)}, y^{(b)}) \sim \mathbb{P}_{XY}, \tilde{y}^{(1)}, \ldots, \tilde{y}^{(b)} \sim \mathbb{P}_Y, \tag{3.10}$$

where $b$ is the number of minibatch samples. Then, based on such defined joint and marginal distributions, we use log-sum-exp techniques to evaluate the lower bound of MI between $\mathbf{X}$ and $\mathbf{Y}$:

$$I_\theta(\mathbf{X}, \mathbf{Y}) \leftarrow \frac{1}{b} \sum_{i=1}^{b} MLP_\theta(x^{(i)}, y^{(i)}) - \log\left(\frac{1}{b} \sum_{i=1}^{b} e^{MLP_\theta(x^{(i)}, \tilde{y}^{(i)})}\right) \tag{3.11}$$

Since it is difficult to accurately measure or calculate the mutual information itself, the MINE method used in this paper theoretically proves that it can be close to the true mutual information, so this paper uses MINE to estimate the mutual information between two variables.

### 3.3 MIXTURE-OF-INFORMATION BOTTLENECK GUIDED FOR DC

#### 3.3.1 MIXTURE-OF-INFORMATION CALCULATION

Assume that various synthetic datasets $[D_1, ..., D_n, ..., D_N]$ with multiple IPC numbers are trained simultaneously. Based on this setting, as shown in Figure 3, we use Eq 3.11 to calculate the mutual information of each synthetic dataset with different IPC numbers compared with others.

To better introduce how to calculate the mixed information bottleneck, we select the synthetic data set with IPC $= 1$ as an example. The loss function of the synthetic dataset (IPC $= 1$) consists of two parts (i.e., matching loss and mixture-of-information bottleneck loss). The former loss is the matching loss function based on the DREAM method, which means matching the intermediate output on the gradient and the features. We define the matching loss to minimize as:

$$\mathcal{L}_{\text{match}} = \frac{\left\|\theta_t^S - \theta_t^D\right\|_2^2}{\left\|\theta_t^D - \theta_0^D\right\|_2^2} + \frac{\left\|F_t^S - F_t^D\right\|_2^2}{\left\|F_t^D - F_0^D\right\|_2^2},$$

where $\theta$, $S$, and $D$ represent the gradients, synthetic dataset, and real dataset, respectively. $F$ means the feature after the convolutional layers.

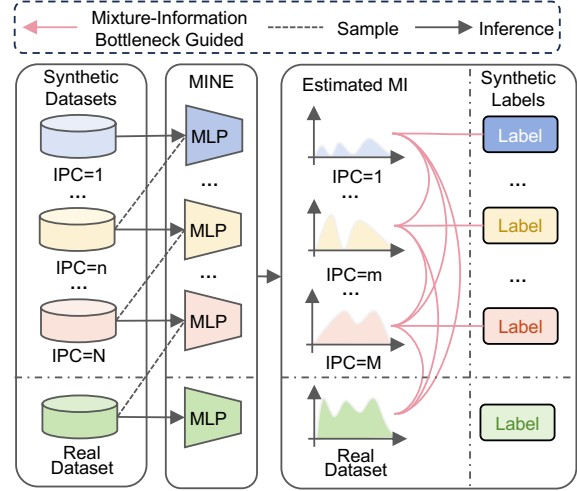

Figure 3: Overview of our Mixture-of-Information Bottleneck Guided method.

The latter is the mixed information bottleneck loss function, which can maximise the mutual information compared with the real data set, IPC = 10, IPC = 50 and other synthetic datasets and decreases the mutual information between labels and itself below a threshold value. We defined the loss function of the mixture-of-information guided loss as follows:

$$\mathcal{L}_{\text{mixture}}^{IPC=n} = I_\theta(\mathbf{X}_{real}, \mathbf{X}_{syn}) - \beta I_\theta(\mathbf{X}_{syn}, \mathbf{Y}) + \alpha \sum_{i=1}^{N} I_\theta(\mathbf{X}_{syn}, \tilde{\mathbf{X}}_{syn}^i). \tag{3.12}$$

Here, $\mathbf{X}_{syn}^i$ represents the other synthetic datasets, $\beta$ and $\alpha$ mean the hyper-parameters in the mixture-of-information bottleneck. The overall loss of the synthetic dataset (IPC=1) can be shown as:

$$\mathcal{L}_{overall} = \mathcal{L}_{match} + \mathcal{L}_{\text{mixture}}^{IPC=n}. \tag{3.13}$$

Table 1: dataset condensation methods comparisons. The settings are the same as previous SoTAs, BPTT Deng & Russakovsky (2022), MTT Cazenavette et al. (2022), and DREAM Liu et al. (2023). Importantly, `MIBDC` can work as an add-on module for SoTA methods.

| | MNIST | | | CIFAR10 | | | CIFAR100 | |
|---|---|---|---|---|---|---|---|---|
| | IPC-1 | IPC-10 | IPC-50 | IPC-1 | IPC-10 | IPC-50 | IPC-1 | IPC-10 |
| Full Set | | 99.6 ± 0.0 | | | 84.8 ± 0.1 | | 56.2 ± 0.3 | |
| DD Wang et al. (2018) | - | 79.5 ± 8.1 | - | - | 36.8 ± 1.2 | - | - | - |
| LD Bohdal et al. (2020) | 60.9 ± 3.2 | 87.3 ± 0.7 | 93.3 ± 0.3 | 25.7 ± 0.7 | 38.3 ± 0.4 | 42.5 ± 0.4 | 11.5 ± 0.4 | - |
| CAFE Wang et al. (2022) | 93.1 ± 0.3 | 97.2 ± 0.2 | 98.6 ± 0.2 | 30.3 ± 1.1 | 46.3 ± 0.6 | 55.5 ± 0.6 | 14.0 ± 0.3 | 31.5 ± 0.2 |
| DM Zhao & Bilen (2023) | 89.7 ± 0.6 | 97.5 ± 0.1 | 98.6 ± 0.1 | 26.0 ± 0.8 | 48.9 ± 0.6 | 63.0 ± 0.4 | 11.4 ± 0.3 | 29.7 ± 0.3 |
| DSA Zhao & Bilen (2021) | 88.7 ± 0.6 | 97.8 ± 0.1 | 99.2 ± 0.1 | 28.8 ± 0.7 | 52.1 ± 0.5 | 60.6 ± 0.5 | 16.8 ± 0.2 | 32.3 ± 0.3 |
| DC Zhao et al. (2021) | 91.7 ± 0.5 | 97.4 ± 0.2 | 98.9 ± 0.2 | 28.3 ± 0.5 | 44.9 ± 0.5 | 53.9 ± 0.5 | 12.8 ± 0.3 | 25.2 ± 0.3 |
| DCC Lee et al. (2022) | - | - | - | 32.9 ± 0.8 | 49.4 ± 0.5 | 61.6 ± 0.4 | 13.3 ± 0.3 | 30.6 ± 0.4 |
| DSAC Lee et al. (2022) | - | - | - | 34.0 ± 0.7 | 54.5 ± 0.5 | 64.2 ± 0.4 | 14.6 ± 0.3 | 33.5 ± 0.3 |
| FRePo Zhou et al. (2022) | 92.4 ± 0.5 | 98.4 ± 0.1 | 98.8 ± 0.1 | 41.3 ± 0.5 | 59.6 ± 0.3 | 63.6 ± 0.2 | 24.8 ± 0.2 | 31.2 ± 0.2 |
| FRePo-w Zhou et al. (2022) | 93.0 ± 0.4 | 98.6 ± 0.1 | 99.2 ± 0.0 | 46.8 ± 0.7 | 65.5 ± 0.4 | 71.7 ± 0.2 | 28.7 ± 0.1 | 42.5 ± 0.2 |
| MTT Cazenavette et al. (2022) | 91.4 ± 0.9 | 97.3 ± 0.1 | 98.5 ± 0.1 | 46.3 ± 0.8 | 65.3 ± 0.7 | 71.6 ± 0.2 | 24.3 ± 0.3 | 40.1 ± 0.4 |
| TESLA Cui et al. (2022) | - | - | - | 48.5 ± 0.8 | 66.4 ± 0.8 | 72.6 ± 0.7 | 24.8 ± 0.4 | 41.7 ± 0.3 |
| MIDD4 Shang et al. (2024) | - | - | - | 51.9 ± 0.3 | 70.8 ± 0.1 | 74.7 ± 0.2 | 31.1 ± 0.4 | 47.4 ± 0.3 |
| DREAM Liu et al. (2023) | 95.7 ± 0.1 | 98.6 ± 0.1 | 99.2 ± 0.1 | 51.1 ± 0.3 | 69.4 ± 0.4 | 74.8 ± 0.1 | 29.5 ± 0.3 | 46.8 ± 0.7 |
| MDC He et al. (2024) | - | - | - | 49.6 ± 0.3 | 66.7 ± 0.4 | 74.5 ± 0.1 | 27.6 ± 0.3 | 41.5 ± 0.7 |
| `MIBDC` | 95.8 ± 0.1 | 98.6 ± 0.1 | 99.2 ± 0.1 | 52.5 ± 0.3 | 70.9 ± 0.1 | 75.2 ± 0.1 | 31.3 ± 0.6 | 47.4 ± 0.3 |
| Δ | (0.1↑) | (0.0-) | (0.0-) | (1.4↑) | (1.5↑) | (0.2↑) | (1.8↑) | (0.6↑) |

# 4 EXPERIMENTS

In this section, we conduct comprehensive experiments to evaluate our proposed method `MIBDC` on three different datasets for the DC task. We designed comprehensive experiments to address the following three Research Questions (RQ):

**RQ1 (Effectiveness of `MIBDC`)**: How does MIB compare to the current state-of-the-art dataset distillation methods in terms of effectiveness?

**RQ2 (Impact of MIB Guidance)**: How does MIB itself affect the loss of the main task? How does MIB impact the contained information of synthetic datasets?

**RQ3 (Impact of MIB Parameters)**: How do the hyperparameters of MIB influence its performance?

We first describe the implementation details of `MIBDC`, and then compare our method with several SoTA DC methods to demonstrate the superiority of our proposed method. Finally, we validate the effectiveness of the MI module (connected with Eq 3.12 and Eq 3.13) by a series of ablation studies.

## 4.1 DATASETS AND IMPLEMENTATION DETAILS

To validate the efficacy and effectiveness of our approach, we implemented our method using PyTorch and compared its performance with various state-of-the-art Dataset Condensation (DC) methods. We conducted all experiments on a Ubuntu server with a 2.9GHz Intel Xeon CPU, 256GB of memory, and an NVIDIA A10 Tensor Core GPU. We use MNIST LeCun et al. (1998), and CIFAR10/100 datasets to conduct our experiments.

**Datasets. MNIST** LeCun et al. (1998) is a dataset for handwritten digits recognition that is widely used for validating image recognition models. It contains 60,000 training images and 10,000 testing images with the size of $28 \times 28$. **CIFAR10/100** Krizhevsky et al. (2009) are two datasets consist of tiny colored natural images with the size of $32 \times 32$ from 10 and 100 categories, respectively. In each dataset, 50,000 images are used for training and 10,000 images for testing.

## 4.2 RQ1: COMPARISON WITH SoTA METHODS

We compare mid with a series of state-of-the-art (SoTA) dataset distillation methods, including DD Wang et al. (2018), LD Bohdal et al. (2020), DC Zhao et al. (2021), DC with Differentiable Siamese Augmentation (DSA) Zhao & Bilen (2021), DC with Distribution Matching (DM) Zhao et al. (2021), CAFE Wang et al. (2022), FRePo Zhou et al. (2022), TESLA Cui et al. (2022), BPTT Deng & Russakovsky (2022), and MTT Cazenavette et al. (2022). Table 1 shows the performance and comparison results of our method across the three datasets. Evaluating the overall performance of MIB on these mainstream dataset extraction benchmarks, it is evident that our approach consistently

outperforms existing SoTA methods, including those based on information gain. For instance, in the scenario of generating 10 images per class, our method achieves the best results across all datasets. Furthermore, when synthesizing 10 images per class using CIFAR100 as a real-world dataset, our method surpasses DREAM and MIDD4 by 1.8% and 0.7%, respectively. Notably, our approach requires only a single training run to generate synthetic datasets of all desired sizes, which significantly enhances the training efficiency for IoT devices of varying scales.

### 4.3 RQ2: THE IMPACT OF MIXTURE-OF-INFORMATION BOTTLENECK GUIDANCE

#### 4.3.1 REGULARIZATION PROPERTY

This section analyzes the changing trend of the dataset distillation loss during training. The loss shown is the sum of the $\mathcal{L}_{\text{match}}$ of IPC=1, IPC=10, and IPC=50, the value of the previous term of Eq 3.11. From Figure 4, we can observe that under the guidance of mixed mutual information, even though the initial loss of MIBDC is not as small as that of DREAM (the randomness introduced by the dataset initialization), as the number of training rounds gradually increases, the final loss also decreases sharply, so that the SoTA performance in Table 1 can be obtained. Therefore, we can conclude that the mixture-of-information guidance loss actually plays a regularization property in the whole DC procedure.

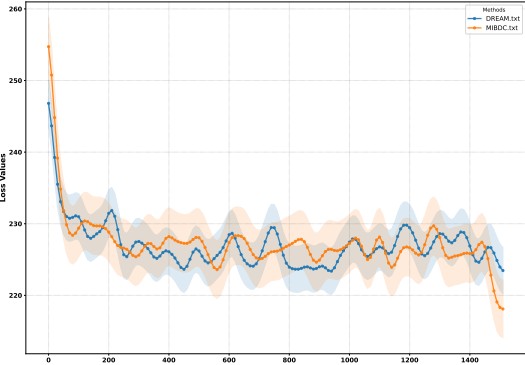

Figure 4: $\mathcal{L}_{\text{match}}$ curves training with/without Mixture-of-information Bottleneck Guidance.

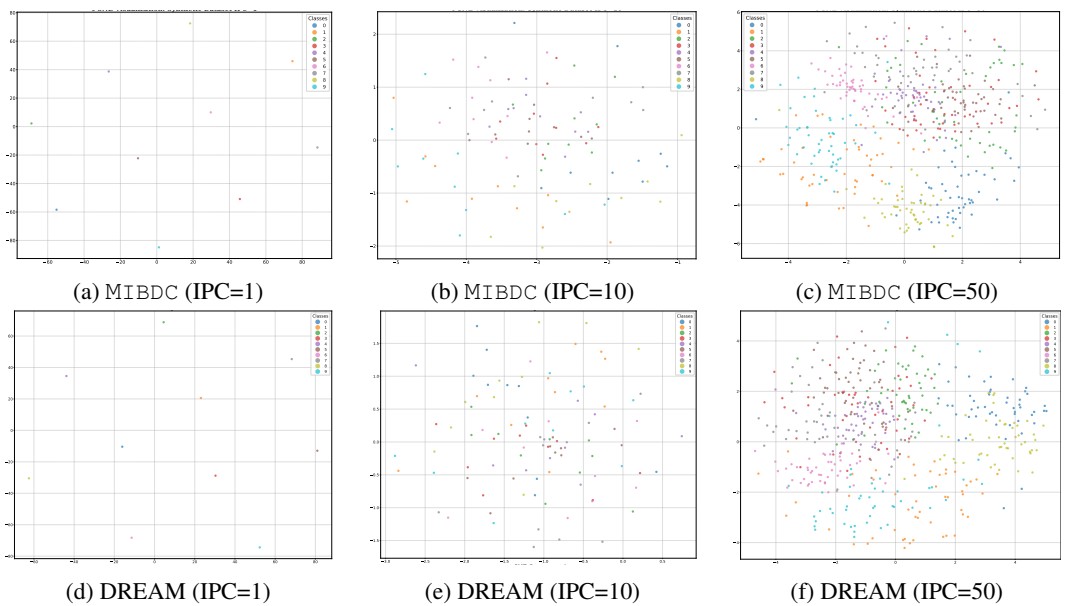

(a) MIBDC (IPC=1)    (b) MIBDC (IPC=10)    (c) MIBDC (IPC=50)

(d) DREAM (IPC=1)    (e) DREAM (IPC=10)    (f) DREAM (IPC=50)

Figure 5: The t-SNE of feature vectors under two dataset condensation method on CIFAR-10.

### 4.3.2 Contained Information Analysis

To describe the detailed information, we use the t-SNE graph to describe the contained information in the synthetic dataset compared to other IPC numbers. As can be seen from the Figure 5, after using MIB guidance, the distances between each class in synthetic datasets with different IPC numbers have been improved (indicated by the scale of the Y-axis), which means that the information contained in the synthetic data sets has been partially improved.

## 4.4 Ablation Study

In this section, we present a series of experiments focused on hyperparameters, using the IPC=1 dataset as a representative example. We examine the effects of parameters $\alpha$ and $\beta$ under three different conditions (0.1, 0.5, 1), with the results summarized in the accompanying Table 2. From the data, we observe that when the value of beta is low, it positively influences the overall performance of the synthetic dataset. However, as beta increases, its impact on performance becomes progressively negative.

Table 2: Hyperparameter tuning results: $\alpha$ and $\beta$ in CIFAR10 with IPC=1

| Alpha ($\alpha$) | Beta ($\beta$) | Accuracy (%) |
|---|---|---|
| 0.1 | 0.1 | 49.1±0.3 |
| 0.1 | 0.5 | 48.3±0.4 |
| 0.1 | 1.0 | 47.6±0.2 |
| 0.5 | 0.1 | 50.7±0.3 |
| 0.5 | 0.5 | 49.5±0.2 |
| 0.5 | 1.0 | 48.7±0.2 |
| 1.0 | 0.1 | **52.5±0.3** |
| 1.0 | 0.5 | 50.9±0.2 |
| 1.0 | 1.0 | 49.6±0.4 |

## 5 Related Work

To address this issue, various methods have been proposed, which can be broadly categorized into two classes: parameter matching methods and distribution matching methods. The parameter matching approach focuses on aligning superior parameters during the DNN training process to optimize the representational capacity of synthetic datasets. For instance, Zhao et al. (2021) used gradient matching between synthetic and real dataset batches with the same DNN. Similarly, Wang et al. (2022) optimized feature maps derived from real datasets, while Liu et al. (2023) employed a combination of techniques to optimize distance, and Shang et al. (2024) used mutual information to measure the distance between synthetic and real datasets. However, since parameters require substantial storage, this approach demands a large amount of memory during training. On the other hand, distribution matching methods aim to enhance the representational capability of synthetic datasets by focusing on the intermediate outputs (i.e., dataset distribution) of DNNs. While both methods can partially improve the representational capacity of synthetic datasets, they inevitably face a trade-off between representational power and storage constraints.

## 6 Conclusion

In this paper, we propose a method named Mixture-of-Information Bottleneck Dataset Condensation (MIBDC) method. The proposed method effectively addresses the limitations of existing data condensation technologies by enhancing both the generalization performance and scalability of synthetic datasets across edge devices. By leveraging information bottlenecks and collaboratively utilizing synthetic datasets with varying images per classses, MIBDC not only improves dataset quality and convergence but also reduces training overheads. The experimental results validate the superiority of MIBDC over state-of-the-art methods, showcasing its potential for efficient deployment in buffer-constrained and heterogeneous edge environments.

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
