# OpenReview forum: "One Training Fits All: Generalized Data Condensation via Mixture-of-Information Bottleneck Guidance"
_ICLR.cc/2025/Conference — Submitted to ICLR 2025_

### Official Review · Reviewer_5KNa · 2024-10-30

**Soundness:** 2
**Presentation:** 2
**Contribution:** 2
**Rating:** 5
**Confidence:** 4

**Summary:**

This paper assumes that the synthetic dataset captures more basic features when ipc is low and more advanced features when ipc is high. So, the authors introduce a Mixture-of-Information Bottleneck method to guide the training, enabling models to effectively learn features at different levels from synthetic datasets with varying IPC values. As a result, the synthetic dataset with low ipc can capture more advanced features. The method is based on calculating mutual information between real and synthetic dataset, as well as between the synthetic dataset and the target.

**Strengths:**

1. The performance improves on CIFAR10/100 datasets.
2. The advanced features in synthetic dataset with low ipc increase in the proposed mixture-information bottleneck-guided method.

**Weaknesses:**

1. The author states it is the first work aiming at optimising the information bottleneck, but there are several papers address the features at different level and the mutual information has also been utilized in published papers.
2. I assume the training cost is large because the author train datasets with all ipcs together. Can you please add the GPU cost.
3. In line 129, the author said there are four datasets, but there are only three datasets in the experiments section and the results table.
4. From line 198 to 200, the text states to minimize I (T, X) and maximize I(T; Y), but equation 3.2 seems to state them in a reverse way.
5. In line 228, the text states to minimize the MI between the Xreal and real label Y, but the second item in equation 3.4 is Xsyn.
6. In line 262, the text is “we can convert the optimization objective Eq 3.9 to the following target”, the equation below is also 3.9. It seems to converts Eq. 3.9 to 3.9.
7. In line 483-484 of conclusion, the author states MIBDC improves dataset quality, but there are no any qualitative results provided in the paper.

**Questions:**

The authors aim to capture more advanced features in the synthetic dataset with low IPC, but what benefits the synthetic dataset with high IPC gains from this method.

---

### Official Review · Reviewer_cw4U · 2024-11-01

**Soundness:** 2
**Presentation:** 3
**Contribution:** 3
**Rating:** 6
**Confidence:** 4

**Summary:**

The paper proposes a new method for data condensation based on information bottleneck. Specifically, the mutual information between synthetic and real datasets is maximized, and the mutual information between synthetic datasets and labels is minimized. Experiments are conducted on MNIST, CIFAR10, and CIFAR100 with similar performance to previous methods.

**Strengths:**

The paper incorporates information bottleneck (IB) for DC, which is interesting and novel, to the best of my knowledge.

**Weaknesses:**

Images Per Class (IPC) may not always be a faithful indicator of the condensation rate as different intact datasets have different sizes. Can you also show it as a percentage of the original dataset size?

The experiment setting could possibly be misleading. Data condensation should be task-agnostic. You use image classification labels during the DC process, and use image classification again on the condensed data for evaluation. However, for a fair evaluation, the tasks before and after DC should be independent. Can you switch the evaluation task, like some other papers did, for a fairer evaluation?

One strength of the proposed method is "our approach requires only a single training run". However, IB itself is time-consuming, and there is no table for the comparison of the running times of different methods.

The improvement is quite marginal (1.8% and 0.7%). More importantly, is this difference statistically significant?

Moreover, the results in Table 1 are obtained (e.g., 52.5) by using the optimal alpha and beta values in Table 2. However, before you perform the hyperparameter tuning, how do you know the best alpha and beta values? This could be a form of data leakage.

The experiment is incomplete. What's the performance on larger, and more realistic datasets like ImageNet?

The paper should be carefully revised as there are multiple typos or formatting issues.

**Questions:**

On page 3, "synthetic datasets with a smaller IPC intend to converge more quickly". Can this be simply because datasets with a smaller IPC have much fewer images, so it is much faster to train? Can you experimentally (or theoretically) justify that there are indeed two classes of features? If not, then this motivation/assumption may not hold true.

On page 4, what is "DD" in "some representative DD methods"? You should give the full form before using the abbreviation form.

Why there is no Y_real in eq. 3.1?

Using information bottleneck for DC is an interesting idea. However, there could be different ways of applying IB on DC. Could you add two more ablation studies, as follows? (1) minimize the mutual information between synthetic and real datasets while maximizing the mutual information between synthetic datasets and labels. (2) In eq. 3.4, use a dummy Y, e.g., its permutation, to examine the effect of this term.

**Details Of Ethics Concerns:**

No ethical issues.

---

### Official Review · Reviewer_v1BH · 2024-11-03

**Soundness:** 2
**Presentation:** 3
**Contribution:** 2
**Rating:** 3
**Confidence:** 3

**Summary:**

The paper introduces Mixture-of-Information Bottleneck Dataset Condensation (MIBDC), a new method designed to improve data condensation technologies used in buffer-constrained scenarios for deep neural network (DNN) training. The authors identify critical limitations in existing approaches, which typically rely on matching intermediate outputs between real and synthetic datasets without enhancing their representational capabilities. MIBDC is claimed to address these issues by leveraging information bottlenecks and varying the number of images per class (IPC) to enhance the generalization and scalability of synthetic datasets across edge devices. The study finds that larger synthetic datasets can guide smaller ones to better convergence and that the collaboration among synthetic datasets with different IPC numbers can improve overall performance. Experimental results on three small/medium datasets demonstrate that MIBDC can achieve good results, enhancing both generalization and scalability while reducing training overheads.

**Strengths:**

The paper leverages the development of Mixture-of-Information Bottleneck Dataset Condensation (MIBDC), which enhances the generalization performance and scalability of synthetic datasets for deep neural network training. By addressing limitations in existing data condensation methods, MIBDC employs information bottlenecks and utilizes synthetic datasets with varying images per class (IPC) to improve representational capabilities and convergence rates. The authors provide evidence that larger synthetic datasets can guide the performance of smaller ones and demonstrate the method's effectiveness by experiments.

**Weaknesses:**

Some aspects can be improved,

1) It seems the proposed method is quite similar to MIM4DD (NeurIPS 2023), and a clearer clarification on how different they are may help.

2) From Table 1, the marginal improvements over MIM4DD (e.g., CIFAR-10: 0.6%, 0.1%, 0.5%; CIFAR-100: 0.2%, 0.0%) should be further contextualized to demonstrate their significance.

3) While the paper reports benefits from using synthetic data, it should consider how advanced deep neural networks (DNNs) achieve high accuracies on CIFAR-10, such as: Rank-1: ViT-H/14, ACC:99.5; Rank-2: DINOv2, ACC: 99.5; etc., more:

    https://paperswithcode.com/sota/image-classification-on-cifar-10. Will these advanced DNNs gain benefits from your approach?

4) A description of the data synthesis approaches used in the study is essential. The quality of synthetic data can vary significantly based on the method applied, which would contextualize the results presented in Table 1.

5) The early convergence of MIBDC (yellow line in Fig. 4) raises questions about whether the blue method would converge better with additional epochs, which warrants further investigation.

6) Figure 5 should include a more detailed interpretation of its results, moving beyond the statement of improvement to explain the underlying reasons for the observed outcomes.

7) Table 2 seems not very valuable - how to set the best hyper-parameters may bring out incremental benefits. But what is its scientific value here?

8) Some citations may be included or compared,

[1] ICLR 2024, Embarrassingly Simple Dataset Distillation

[2] https://arxiv.org/abs/2406.01112

[3] https://arxiv.org/abs/2408.14506

[4] DREAM++ https://arxiv.org/abs/2310.15052

[5} You Only Condense Once, https://proceedings.neurips.cc/paper_files/paper/2023/hash/7bdd36a198a8408f444834039b09f518-Abstract-Conference.html

[6] https://openaccess.thecvf.com/content/ICCV2023/html/Liu_Few-Shot_Dataset_Distillation_via_Translative_Pre-Training_ICCV_2023_paper.html

[7] https://openaccess.thecvf.com/content/CVPR2024/html/Gu_Efficient_Dataset_Distillation_via_Minimax_Diffusion_CVPR_2024_paper.html

[8] https://openaccess.thecvf.com/content/CVPR2024/html/Shao_Generalized_Large-Scale_Data_Condensation_via_Various_Backbone_and_Statistical_Matching_CVPR_2024_paper.html

etc..

**Questions:**

No extra questions. See comments above.

---

### Official Review · Reviewer_AsEW · 2024-11-04

**Soundness:** 3
**Presentation:** 3
**Contribution:** 3
**Rating:** 6
**Confidence:** 3

**Summary:**

The paper proposes mixture-of-information bottleneck as guidance to perform data condensation. In particular, the authors propose that  a smaller synthetic dataset can converge more effectively under the guidance of larger dataset and synthetic dataset with different IPC numbers can enhance the generalizability of dataset condenstation.

**Strengths:**

1. The paper is well organized and written so that it is easy to follow and understand.
2. The motivation and idea are clearly presented and explained.
3. The experiment is solid.

**Weaknesses:**

1. In line 140-142,  what are low-density and high-density information?
2. Line 178, DD is a typo?  Should be DC?

**Questions:**

see weakness

---

### Meta-Review · Area_Chair_1PVf · 2024-12-11

**Metareview:**

This paper proposes a dataset distillation method, MIBDC, which leverages information bottlenecks to improve generalization and scalability. However, the authors have completely ignored the rebuttal process, failing to respond to a single question or concern raised by the reviewers. Besides, half of the reviewers have provided negative ratings, showing concerns regarding the quality of the submission.

**Additional Comments On Reviewer Discussion:**

The authors did not reply to the reviews.

---

### Decision · Program_Chairs · 2025-01-22

Reject